# VideoUFO: A Million-Scale User-Focused Dataset for Text-to-Video Generation

**Wenhao Wang**
University of Technology Sydney
wangwenhao0716@gmail.com

**Yi Yang**[*]
Zhejiang University
yangyics@zju.edu.cn

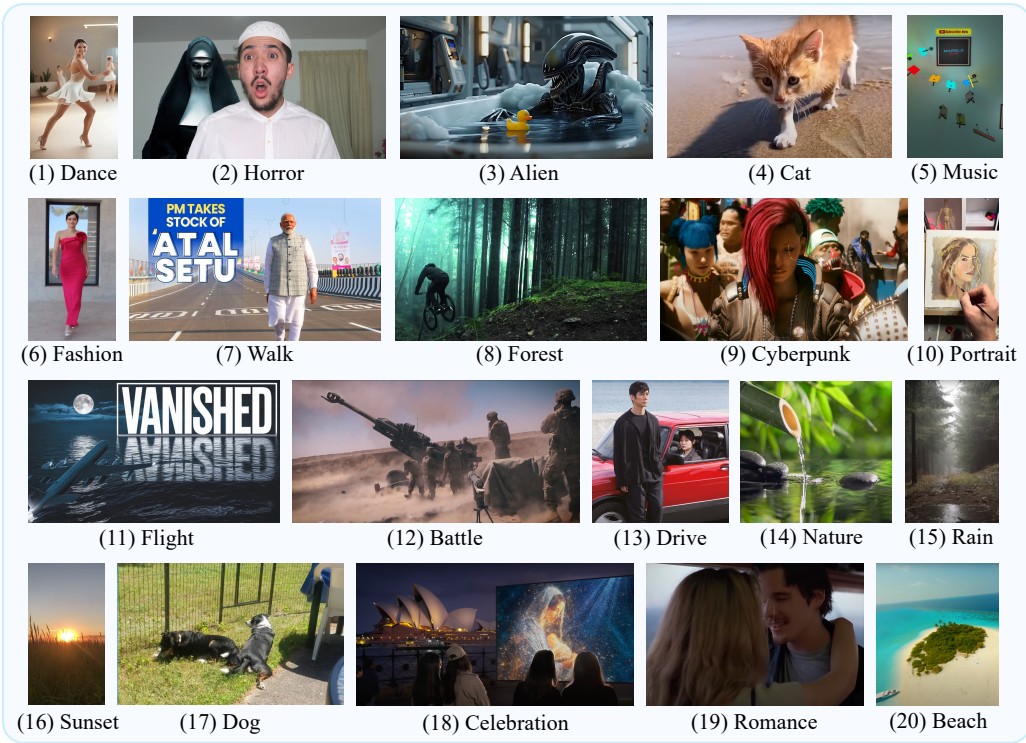

Figure 1: **VideoUFO** is the first dataset curated in alignment with real-world users' focused topics for text-to-video generation. Specifically, the dataset comprises over 1.09 million video clips spanning 1, 291 topics. Here, we select the top 20 most popular topics for illustration. Researchers can use our VideoUFO to train or fine-tune their text-to-video generative models to better meet users' needs.

## Abstract

Text-to-video generative models convert textual prompts into dynamic visual content, offering wide-ranging applications in film production, gaming, and education. However, their real-world performance often falls short of user expectations. One key reason is that these models have not been trained on videos related to some topics users want to create. In this paper, we propose **VideoUFO**, the first **Video** dataset specifically curated to align with **U**sers' **FO**cus in real-world scenarios. Beyond this, our VideoUFO also features: (1) minimal (0.29%) overlap with existing

---

[*]Corresponding Author.

39th Conference on Neural Information Processing Systems (NeurIPS 2025) Track on Datasets and Benchmarks.

video datasets, and (2) videos searched exclusively via YouTube's official API under the Creative Commons license. These two attributes provide future researchers with greater freedom to broaden their training sources. The VideoUFO comprises over 1.09 million video clips, each paired with both a brief and a detailed caption (description). Specifically, through clustering, we first identify $1,291$ user-focused topics from the million-scale real text-to-video prompt dataset, VidProM. Then, we use these topics to retrieve videos from YouTube, split the retrieved videos into clips, and generate both brief and detailed captions for each clip. After verifying the clips with specified topics, we are left with about 1.09 million video clips. Our experiments reveal that (1) current 16 text-to-video models do not achieve consistent performance across all user-focused topics; and (2) a simple model trained on VideoUFO outperforms others on worst-performing topics. The dataset and code are publicly available here and here under the CC BY 4.0 License.

# 1 Introduction

Text-to-video generation aims to convert textual descriptions into dynamic visual content. Its applications are extensive and transformative, covering areas from creative media [1] and entertainment [2] to practical domains such as education [3], advertising [4], and assistance [5].

Despite their popularity and usefulness, current text-to-video models often fail to meet users' expectations in real-world applications. For example, when we ask Sora [6] to generate a video using the prompt "*A firefly is glowing on a grass's leaf on a serene summer night*", it fails to capture the concept of a glowing firefly while successfully generating grass and a summer night, as shown in Fig. 2 (a). From the

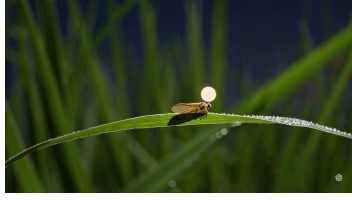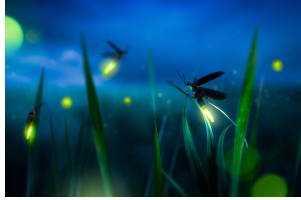

(a) Video generated by Sora          (b) Real video

Figure 2: The *glowing firefly*: (a) generated by Sora [6] and (b) captured in a real video. The generated firefly is noticeably different from its real-life counterpart and thus unsatisfying. We attribute this primarily to a lack of exposure to such topics.

data perspective, we infer this is mainly because Sora [6] has not been trained on *firefly*-related topics, while it has been trained on *grass* and *night*. Furthermore, if Sora [6] has seen the video shown in Fig. 2 (b), it will understand what a glowing firefly should look like.

To this end, this paper presents **VideoUFO**, the first **Video** dataset curated specifically based on real-world **U**sers' **FO**cus in text-to-video generation. Such a dataset can help improve the alignment between text-to-video models and actual user needs. Specifically, we focus on: (**1**) curating VideoUFO by analyzing real users' interests and scraping relevant videos; (**2**) comparing VideoUFO with other datasets to highlight differences; and (**3**) showing how VideoUFO benefits video generation.

**The first dataset that aligns with real-world users' focus in text-to-video generation.** Our key idea is to analyze user-focused topics from user-provided prompts and then search for videos related to these topics. As shown in Fig. 1, the resulted VideoUFO comprises more than **1.09** million video clips spanning **1,291** user-focused topics. Specifically, we initiate by analyzing user-focused topics, which involves embedding all 1.67 million user-provided prompts from VidProM [7], clustering these embeddings using K-means, and generating a topic for each cluster with GPT-4o [8], followed by combining similar topics. After obtaining these topics, we (1) search for these topics on YouTube, (2) segment the fetched videos into multiple semantically consistent short clips, (3) generate both brief and detailed captions for each clip, (4) filter out clips that do not contain the specific topics, and (5) assign each clip video quality scores that align with human perception. Note that although our current VideoUFO comprises about **one million** videos, it can be easily scaled up to **ten million** or more by sourcing additional videos for each topic. The VideoUFO can also be easily extended to the **image-to-video** domain by using the text and image prompts in TIP-I2V [9].

**Differences between VideoUFO and other recent video datasets.** Recently, several video datasets have been released, including OpenVid-1M [10], HD-VILA-100M [11], InternVid [12], Koala-36M [13], LVD-2M [14], MiraData [15], Panda-70M [16], VidGen-1M [17], and WebVid-10M [18].

While inheriting their excellent attributes – such as large scale, accurate captions, and high resolution – our VideoUFO explores some novel directions: (1) Guided by real user focus. Specifically, whereas recent datasets are typically gathered from open-domain sources, we concentrate on topics that are focused by text-to-video users. This feature enables text-to-video models trained on our dataset to better cater to users, while avoiding unnecessary expansion of the dataset and wasting resources. (2) Introducing new data. Although these recent papers claim to contribute new datasets, most of them primarily introduce a new data pipeline – that is, they reprocess HD-VILA-100M [11] and are subsets of it. While their contributions are useful and meaningful, in theory, a generative model that has already been fully fitted on HD-VILA-100M [11] would not gain new video knowledge from them. In contrast, we collect new data from YouTube, with only $0.29\%$ of the videos overlapping with existing datasets. (3) Data compliance. When curating VideoUFO, we retrieve videos using YouTube's official API and select only those with a Creative Commons license. In contrast, most recent datasets do not explicitly address the regulatory compliance of their data collection process. This feature grants researchers greater flexibility in using our data.

**Benchmarking current text-to-video models on user-focused topics and demonstrating the effectiveness of our VideoUFO.** We present a new benchmark to quantify the performance of text-to-video models on user-focused topics. Specifically, our process involves: (1) selecting 10 user-provided prompts per topic; (2) generating a video for each prompt; (3) using a multimodal large language model to describe each video; and (4) calculating the similarity between the generated descriptions and the original prompts. For each topic, we calculate the average similarity between the 10 prompts and their corresponding descriptions. By sorting these averages, we can identify the worst-performing and best-performing topics for each model. Using this benchmark, we evaluate the performance of current text-to-video models and a newly trained text-to-video model on our VideoUFO. Experimental results indicate that (1) current 16 text-to-video models have some poor-performing topics, and (2) our model achieves the highest similarities on worst-performing topics while maintaining performance on the best-performing ones.

In conclusion, our key contributions are as follows:

1. We present VideoUFO, the first video dataset curated based on the focus of real text-to-video users. This dataset comprises over $1.09$ million clips spanning $1,291$ user-focused topics.

2. We compare VideoUFO with recent video datasets, highlighting their differences in both fundamental attributes and topics coverage, thereby emphasizing the necessity of our dataset. We also follow best practices in their curation processes to ensure the quality of our dataset.

3. We evaluate current text-to-video models on user-focused topics and observe that a simple model trained on our VideoUFO outperforms competing models on worst-performing topics.

## 2 Related Works

**Text-to-Video Generation.** The introduction of Sora [6] has sparked significant research interest in text-to-video generation. Commercial models such as Movie Gen [19], Veo [20], Kling [21], Pixel-Dance [22], and Seaweed [22] demonstrate strong performance and are increasingly being integrated into various industries. Meanwhile, open-source models like HunyuanVideo [23], LTX-Video [24], CogVideoX [25], Mochi-1 [26], and Pyramidal [27] empower researchers to experiment, customize, and enhance existing frameworks. Although they are powerful, their training data determines the upper limit of generated videos' quality. This paper proposes a dataset, VideoUFO, which has the potential to help these models better cater to users' preferences in real-world applications.

**Text-Video Datasets.** A text-video dataset consists of video clips paired with corresponding textual descriptions or captions. Many text-video datasets, such as OpenVid-1M [10], HD-VILA-100M [11], InternVid [12], Koala-36M [13], LVD-2M [14], MiraData [15], Panda-70M [16], VidGen-1M [17], and WebVid-10M [18], have been proposed to advance multimodal understanding and generation. However, these datasets are not designed to align with the preferences of text-to-video users, creating a gap between training data topics and real-world applications. In contrast, our dataset VideoUFO is collected from a user-focus perspective. To better cater users, future researchers can fine-tune text-to-video models with the VideoUFO or integrate it with existing datasets to develop new models.

**Preference Alignment in Diffusion Models.** While diffusion models have demonstrated impressive capabilities, there is an increasing demand to ensure their outputs align with human preferences.

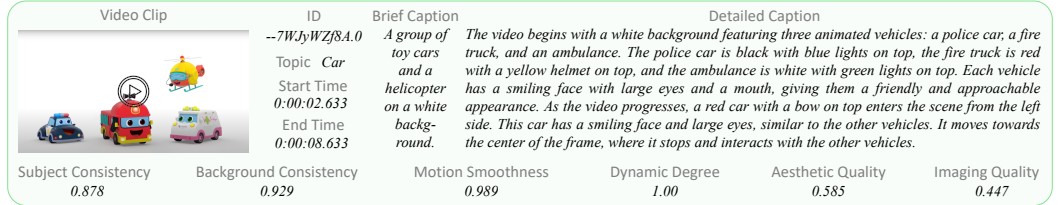

| Video Clip | ID | Brief Caption | Detailed Caption |
|---|---|---|---|
| | --7WJyWZf8A.0 | A group of toy cars and a helicopter on a white backg-round. | The video begins with a white background featuring three animated vehicles: a police car, a fire truck, and an ambulance. The police car is black with blue lights on top, the fire truck is red with a yellow helmet on top, and the ambulance is white with green lights on top. Each vehicle has a smiling face with large eyes and a mouth, giving them a friendly and approachable appearance. As the video progresses, a red car with a bow on top enters the scene from the left side. This car has a smiling face and large eyes, similar to the other vehicles. It moves towards the center of the frame, where it stops and interacts with the other vehicles. |
| | Topic Car | | |
| | Start Time 0:00:02.633 | | |
| | End Time 0:00:08.633 | | |

| Subject Consistency | Background Consistency | Motion Smoothness | Dynamic Degree | Aesthetic Quality | Imaging Quality |
|---|---|---|---|---|---|
| 0.878 | 0.929 | 0.989 | 1.00 | 0.585 | 0.447 |

Figure 3: Each data point in our **VideoUFO** includes a video clip, an ID, a topic, start and end times, a brief caption, and a detailed caption. Beyond that, we evaluate each clip with six different video quality scores from VBench [28].

Direct Preference Optimization (DPO) [29], originally proposed to align large language models (LLMs) with human preferences, can also be effectively applied to diffusion models. For example, DiffusionDPO [30] fine-tunes text-to-image models using human comparison data to improve image generation quality. Additionally, VideoDPO [31] introduces a comprehensive preference scoring system that evaluates both visual quality and semantic alignment to enhance text-to-video generation. Unlike other approaches that align text-to-image or text-to-video models with attributes such as aesthetics, motion, consistency, and visual appeal, our goal is to improve the performance of text-to-video models on real-world users' focused topics.

## 3 Curating VideoUFO

Fig. 3 illustrates a data point from our million-scale VideoUFO, which comprises: (1) a video clip along with its ID; (2) the topic of the clip; (3) the start and end times within its original video; (4) the corresponding brief and detailed captions; and (5) the quality of the clip. The following provides an explanation of how we curate the VideoUFO.

**Analyzing text-to-video users' focused topics.** We aim to analyze the topics that real users focus on or prefer when generating videos from texts. Here, we use VidProM [7] as it is the only publicly available, million-scale text-to-video prompt dataset written by real users. First, we embed all 1.67 million prompts from VidProM into 384-dimensional vectors using `SentenceTransformers` [33]. Next, we cluster these vectors with K-means. Note that here we pre-set the number of clusters to a relatively large value, *i.e.*, 2,000, and merge similar clusters in the next step. Finally, for each cluster, we ask GPT-4o [8] to conclude a topic $(1 \sim 2$ words). The prompt is shown in the Appendix (Section A). After merging singular and plural forms, restoring verbs to their base forms, removing duplicates, and manually verifying the topics – removing overly broad ones such as '*animation*', '*scene*', '*movement*', and '*film*' – we finally obtained **1,291** topics. The semantic distribution of these topics is shown in Fig. 4.

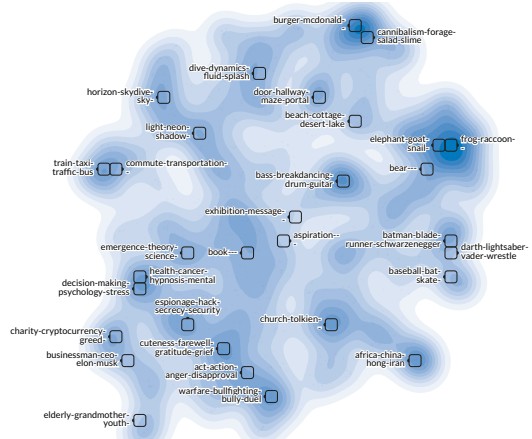

Figure 4: The semantic distribution of users' focused topics. It is visualized by WIZMAP [32]. Please 🔍 zoom in or visit here to see the details.

**Collecting videos from YouTube.** For each topic obtained, we use the official YouTube API to search for videos. Specifically, for each topic, we search for approximately 500 videos by relevance, with each video meeting the following criteria: (**1**) it is shorter than 4 minutes, (**2**) it has a resolution of 720p or higher, and (**3**) it is licensed under Creative Commons. These requirements ensure that the collected videos are **suitable** and **freely usable** for training video generation models. In the end, we obtain 586,490 videos from YouTube. We compare the YouTube IDs of these videos with those in existing datasets, including OpenVid-1M [10], HD-VILA-100M [11], InternVid [12], Koala-36M [13], LVD-2M [14], MiraData [15], Panda-70M [16], VidGen-1M [17], and WebVid-10M [18], and find that only 1,675 IDs (0.29%) are already present in these datasets. This suggests that our

Table 1: The comparison between the VideoUFO with recent video datasets based on their **fundamental attributes**. Unlike previous collections, our VideoUFO is derived directly from real user-focused topics and also offers a more flexible license, while remaining comparable to these datasets in other aspects. "#" and "‾" are abbreviations for "numbers" and "average", respectively.

| Dataset | #Vid. | Len.‾ | Words‾ | Resolution | Domain | #Topic | License |
|---|---|---|---|---|---|---|---|
| WebVid-10M [18] | 10M | 18.0s | 14.2 | <360p | Open | 1,000 | Retracted |
| HD-VILA-100M [11] | 103M | 13.4s | 32.5 | 720p | Open | 648 | R-UDA |
| InternVid [16] | 234M | 11.7s | 17.6 | 720p | Open | 1,051 | Apache 2.0 |
| Panda-70M [16] | 70M | 8.5s | 13.2 | 720p | Open | 719 | R-UDA |
| LVD-2M [14] | 2M | 20.2s | 88.7 | Diverse | Open | 814 | R-UDA |
| MiraData [15] | 0.33M | 72.1s | 318.0 | 720p | Open | 639 | GPL 3.0 |
| Koala-36M [13] | 36M | 17.2s | 202.1 | 720p | Open | 724 | R-UDA |
| VidGen-1M [17] | 1M | 10.6s | 89.3 | 720p | Open | 835 | R-UDA |
| OpenVid-1M [10] | 1M | 7.2s | 127.3 | Diverse | Open | 671 | R-UDA |
| ChronoMagic-Pro [36] | 460K | 234s | 1794.0 | 720p | Time | 792 | CC BY |
| **VideoUFO** | 1M | 12.6s | 155.5 | 720p | **Users** | **1,291** | **CC BY** |

VideoUFO introduces novel information or knowledge, which can serve to **expand the range of existing training sources**.

**Splitting videos and generating captions.** After obtaining the videos, we segment them into multiple semantics-consistent short clips following the steps in curating Panda-70M [16]. The process includes *shot boundary detection*, *stitching*, and *video splitting*, producing a total of $3,181,873$ clips. To facilitate the training of text-to-video models on our VideoUFO, we generate both brief and detailed captions for each video clip. For the brief captions, we utilize the video captioning model provided by Panda-70M [16]. For the detailed captions, we adopt the pipeline used in `Open-Sora-Plan` v1.3.0 [34], which employs `QWen2-VL-7B` [35] for video annotation. In the Appendix (Fig. 10 (a) and (b)), we present the statistical distribution of the lengths of brief and detailed captions, respectively.

**Verifying clips.** We notice that, although the videos are searched by topics, not every clip contains our intended topics. A straightforward solution is to use GPT-4o [8] to verify whether a given clip corresponds to a specific topic. However, while effective, this approach would be prohibitively expensive for verifying more than 3 million video clips. As an alternative, we use the **detailed caption** (instead of the original clip) and feed it into **GPT-4o mini** (instead of GPT-4o) to verify whether it belongs to a specific topic. Since (1) GPT-4o mini and GPT-4o have similar capabilities in basic language understanding, and (2) the video understanding model provides detailed descriptions of the videos, we effectively complete the final verification step at a significantly lower cost. After verifying, there are $1,091,712$ remaining clips. In the Appendix (Fig. 10 (c)), we present the statistical distribution of clips duration.

**Video quality assessment.** To further support research in text-to-video generation, we evaluate the quality of each clip in VideoUFO. Specifically, we adopt six different video quality assessment metrics from VBench [28], which automatically assess videos and align well with human perception. The evaluation dimensions are: (**1**) *subject consistency*: assesses whether the main subject's identity and appearance remain consistent throughout the video; (**2**) *background consistency*: evaluates the temporal stability and uniformity of the video's background; (**3**) *motion smoothness*: measures the continuity and fluidity of movements within the video; (**4**) *dynamic degree*: assesses the level of activity and variation in motion present in the video; (**5**) *aesthetic quality*: judges the visual appeal and attractiveness of the video; and (**6**) *imaging quality*: examines the clarity, brightness, and color accuracy of the video. The statistical distributions of scores assessed by these six metrics are shown in the Appendix (Fig. 11).

**Extension.** Future researchers can easily extend our VideoUFO in three aspects: (**1**) **Scaling up.** Although our VideoUFO has already reached a million-scale level, future researchers may still want to scale it up to 10 million or even 100 million. They can easily achieve this by leveraging our extracted topics to search more videos on platforms such as YouTube and TikTok. (**2**) **New focusing.** When we curate VideoUFO, the only available text-to-video prompt dataset is VidProM [7], which is used to analyze user focus and preferences. In the future, user focus may change, and other text-to-video prompt datasets may emerge for analyzing it. Future researchers can easily use our topic extraction pipeline to study these new focuses. (**3**) **Image-to-video.** Our current focus is text-to-video, which is the most common approach in the video generation community. Meanwhile, we also notice that

image-to-video is gaining popularity. Future researchers can use text and image prompts, such as those in TIP-I2V [9], to analyze the focus of image-to-video users and collect corresponding datasets.

# 4 Comparison with Other Video Datasets

This section compares the proposed VideoUFO with other recent video datasets in terms of *fundamental attributes* and *topics coverage*. These differences underscore the necessity of introducing VideoUFO for text-to-video generation.

## 4.1 Fundamental Attributes

From the Table 1, we draw three conclusions:

- VideoUFO is collected in line with real text-to-video **users' focus** or preferences. In contrast to VideoUFO, other datasets collect videos from the open domain, which may not cover the topics users focus on, and text-to-video models trained on them may fail to meet users' needs.

- VideoUFO is released under a more **flexible license** and introduces **new data**. Specifically, we search for videos on YouTube by ourselves and only select those with a Creative Commons license. In contrast, most recent datasets (including Panda-70M [16], LVD-2M [14], Koala-36M [13], VidGen-1M [17], and OpenVid-1M [10]) directly source data from HD-VILA-100M [11]. As a result: (1) they do not contribute new data but rather introduce a new data processing pipeline; (2) they must adhere to the same license as HD-VILA-100M, *i.e.*, the Research Use of Data Agreement (R-UDA), which restricts commercial use. In addition, one of the most widely used datasets, WebVid-10M [18], has been retracted due to potential copyright infringement.

- VideoUFO **inherits best practices** from other curated datasets. We observe that recent datasets (1) feature detailed captions generated by multimodal large language models (MLLMs), (2) contain millions of video clips, and (3) are high-resolution (*i.e.*, 720p). Therefore, to provide a large-scale and high-quality resource, our VideoUFO inherits these features.

## 4.2 Topics Coverage

This section analyzes the differences in topics coverage between the VideoUFO and existing datasets.

**Calculation process.** We calculate the number of user-focused topics covered by the existing datasets as follows:

- Extract topics of recent video datasets. We repeat VidProM's topic extraction process (Step 1 in Section 3) on the video datasets listed in Table 1. The number of topics extracted for each dataset is shown in Table 1 (#Topics).

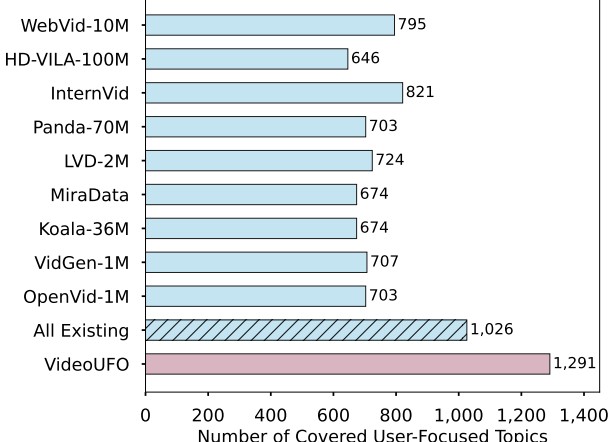

Figure 5: The number of user-focused topics covered by recent video datasets. None successfully includes all user-focused topics.

- Match the extracted topics with users' focused ones. We observe that the same or similar topic may be described using different words. For example, both *cathedral* and *church* refer to a similar topic. Therefore, when comparing user-focused topics with those from each existing dataset, we choose semantic matching rather than a word-to-word approach. Specifically, (1) we first use `SentenceTransformers` [33] to embed the two lists of topics; and (2) then for each user-focused topic, if there exists a topic in the existing dataset with a cosine similarity greater than 0.6, we consider that user-focused topic to be covered by the existing dataset. Note that the threshold of 0.6 is an empirical value, as we observe that most similar topics exhibit a cosine similarity greater than 0.6.

**Observations.** The experimental results in Fig. 5 shows:

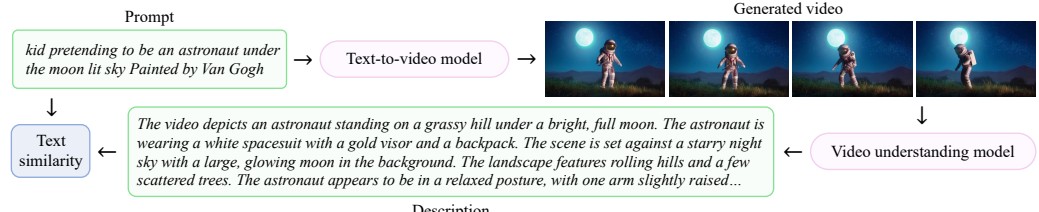

Figure 6: The calculation process of BenchUFO. It is designed to evaluate whether a text-to-video model can effectively generate videos that contain user-focused topics. It comprises 791 concrete noun topics, each paired with 10 real-world user-provided prompts.

- None of the existing datasets cover all user-focused topics in VideoUFO. Specifically, the most comprehensive dataset, InternVid [12], covers 821 topics, accounting for 63.6% of all user-focused topics. Furthermore, if we combine all existing datasets in Table 1 into the "largest" one, the coverage of user-focused topics will increase to 79.5%. However, this will lead to a higher computational burden when training text-to-video models.

- Interestingly, a dataset may cover more user-focused topics than the actual number of topics it contains. For instance, OpenVid-1M [10] contains 671 topics but covers 703 user-focused ones. This is reasonable because two topics in our dataset, VideoUFO, may be similar and thus mapped to a single topic in OpenVid-1M [10]. Nevertheless, this does **not** compromise the quality of our VideoUFO since it only leads to some relative duplicated topics, while the videos in VideoUFO still accurately reflect real users' focus and interests.

**Limitation.** The above topic analysis is based directly on the captions/descriptions provided by each recent video dataset. This is because downloading and re-captioning all these videos is prohibitively expensive in terms of network bandwidth and computation. The quality of these captions/descriptions may affect the number of extracted topics, as some may not be informative. To address this limitation, in the next section, we demonstrate that text-to-video models trained on these datasets fail to generate satisfactory videos for some topics.

## 5 VideoUFO Benefits Video Generation

### 5.1 Benchmark

This section details the proposed benchmark, BenchUFO, for evaluating text-to-video models' performance on user-focused topics. We first introduce the design of the benchmark and then explain why it is reasonable.

**Construction.** As shown in Fig. 6, the calculation process of our BenchUFO includes: **(1) Selecting prompts.** First, we select 791 concrete nouns from $1,291$ user-focused topics. (Please refer to the next section for the rationale behind selecting concrete nouns.) Then, for each chosen topic, we randomly select 10 text prompts from VidProM [7]. All these prompts constitute the prompt set for our benchmark. **(2) Generating videos.** For each prompt, we use a text-to-video model to generate a corresponding video. **(3) Describing videos.** For each video, we use a multimodal large language model (here, we choose QWen2-VL-7B [35]) to understand and describe it. The instruction prompt is provided in the Appendix (Section B). **(4) Calculating similarity.** We use a sentence embedding model (in this case, SentenceTransformers [33]) to encode both the input prompt and the output description, and then compute the cosine similarity between them. We calculate the average similarity across 10 prompts for each topic. A higher similarity score indicates better performance on that specific topic. Our benchmark computes and considers the 10/50 worst-performing and 10/50 best-performing topics.

**Justification.** This section provides justification for the proposed BenchUFO from three perspectives:

- Why choose concrete nouns? This is due to the inconsistency between the input prompt and the output description when using abstract nouns. For instance, for the abstract noun topic *"freedom"* and the prompt *"In a freedom world, ideal and love both exist"*, a model may generate videos containing *"dove"*, *"star"*, or *"heart"*, and the video understanding model will summarize them as *"a dove is playing..."*, *"a star is shining..."*, and *"a heart is beating..."*. However, the embeddings of

these descriptions are not expected to be close to those of the given prompt. This might create a false impression that the model cannot effectively generate content for that topic.

- Why use video understanding model reasonable? This is because the video understanding model has advanced significantly with the advent of large language models, and it is reasonable to assume that these models have been exposed to images/videos containing user-focused topics. Here, we choose `QWen2-VL-7B` [35] as the video understanding model. Trained on 800 billion tokens of visual-related data and 600 billion tokens of text data, it is expected to accurately comprehend these user-focused topics and faithfully summarize a video.

- Why not use established benchmarks? This is because the existing benchmarks fail to reflect real-world scenarios. Specifically, we note that there exists a well-established benchmark, VBench [28], whose "*object class*" and "*multiple objects*" dimensions are similar to our BenchUFO. However, the prompts in VBench have two main drawbacks: **(1)** They cover only a limited number of topics/objects – specifically, just 79. **(2)** They focus exclusively on common topics/objects (e.g., "*bicycle*", "*car*", and "*airplane*"), while many topics that users care about, such as "*forest*", "*sunset*", and "*beach*", are missing.

## 5.2 Observations

In this section, we evaluate current text-to-video models on this new benchmark and show that, with the help of VideoUFO, we achieve state-of-the-art performance. The quantitative and qualitative results are shown in Table 7 and Fig. 8, respectively. We observe that:

- Current text-to-video models do not consistently perform well across all user-focused topics. Specifically, there is a score difference ranging from 0.233 to 0.314 between the top-10 and low-10 topics. These models may not effectively understand topics such as "*giant squid*", "*animal cell*", "*Van Gogh*", and "*ancient Egyptian*" due to insufficient training on such videos.

- Current text-to-video models show a certain degree of consistency in their best-performing topics. We discover that most text-to-video

Figure 7: The performance of both publicly available text-to-video models and our trained models on the proposed BenchUFO. The publicly available models are trained on various datasets, including both public and private ones. MVDiT [10] is trained on VidGen [17], OpenVid [10], and VideoUFO, respectively. "Low/Top $N$" denotes the average score of the worst/best-performing $N$ topics.

| Models | Low 10 | Low 50 | Top 50 | Top 10 |
|---|---|---|---|---|
| Mira [37] | 0.236 | 0.282 | 0.508 | 0.550 |
| Show-1 [38] | 0.266 | 0.303 | 0.524 | 0.564 |
| LTX-Video [24] | 0.268 | 0.310 | 0.532 | 0.574 |
| Open-Sora-Plan [34] | 0.314 | 0.361 | 0.559 | 0.598 |
| TF-T2V [39] | 0.316 | 0.359 | 0.560 | 0.595 |
| Mochi-1 [26] | 0.323 | 0.367 | 0.580 | 0.616 |
| HiGen [40] | 0.352 | 0.394 | 0.589 | 0.625 |
| Open-Sora [41] | 0.363 | 0.409 | 0.601 | 0.639 |
| Pika [42] | 0.365 | 0.404 | 0.583 | 0.619 |
| RepVideo [43] | 0.368 | 0.402 | 0.589 | 0.619 |
| T2V-Zero [44] | 0.375 | 0.410 | 0.586 | 0.616 |
| CogVideoX [25] | 0.383 | 0.419 | 0.601 | 0.629 |
| Latte-1 [45] | 0.384 | 0.421 | 0.592 | 0.636 |
| HunyuanVideo [23] | 0.388 | 0.427 | 0.612 | 0.645 |
| LaVie [46] | 0.399 | 0.426 | 0.595 | 0.632 |
| Pyramidal [27] | 0.400 | 0.433 | 0.606 | 0.647 |
| MVDiT-VidGen [17] | 0.382 | 0.426 | 0.594 | 0.626 |
| MVDiT-OpenVid [10] | 0.401 | 0.437 | 0.609 | 0.645 |
| **MVDiT-VideoUFO** | **0.442** | **0.465** | **0.619** | **0.651** |

models excel at generating videos on animal-related topics, such as '*seagull*', '*panda*', '*dolphin*', '*camel*', and '*owl*'. We infer that this is partly due to a bias towards animals in current video datasets.

- The proposed VideoUFO helps reduce the gap between the worst-performing and best-performing topics. To demonstrate the effectiveness of the proposed VideoUFO, we train an MVDiT [10] solely on VideoUFO. We find that the trained model achieves the highest low-10 scores (a +4.2% improvement compared to the current state-of-the-art) while maintaining performance on the top-10 topics. Visually, the trained model successfully generates videos on topics that other models previously could not.

- The proposed VideoUFO outperforms other similar-scale datasets, such as OpenVid [10] and VidGen [17]. To demonstrate that the improvement originates from our VideoUFO rather than

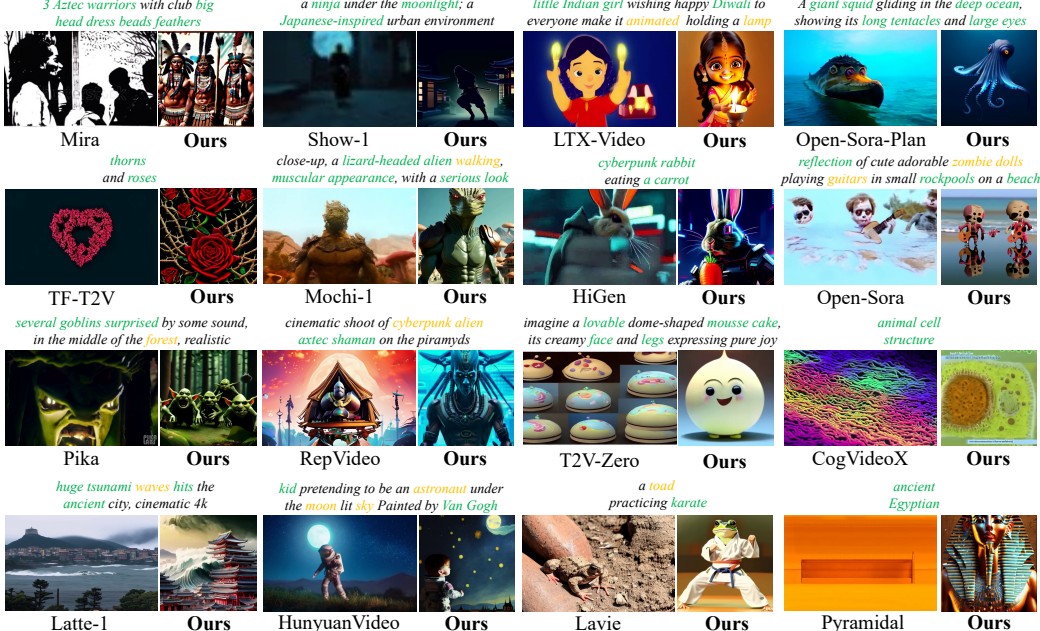

Figure 8: Visual comparisons between our approach (MVDiT-VideoUFO) and other text-to-video models. The model trained on VideoUFO outperforms the alternatives in generating user-focused topics. In the prompts, concepts in green indicate those successfully generated **solely** by our method, whereas those in yellow denote concepts successfully generated by both our model and the competing models. All generated videos are provided in the supplementary materials.

from the MVDiT architecture [10], we replace our VideoUFO with OpenVid [10] and VidGen [17]. We find that when using the popular OpenVid [10], the low-10 scores are similar to previous state-of-the-art models (0.401 vs. 0.400). Furthermore, when training on VidGen [17], the performance is even lower.

**Limitation.** Due to limited computational resources, when validating the effectiveness of our dataset, we currently follow the video generation method from the academic community introduced by [10], which already requires 32 A100 GPUs. However, the quality of videos generated by this baseline remains limited. In the future, industry researchers could incorporate our dataset into their large-scale foundational model training pipelines to further explore its potential.

## 6 Ablation Studies on Curation of VideoUFO

This section performs ablation studies to investigate key components in curating VideoUFO, *i.e.*, examining (1) the impact of the number of clusters in user-focused topic analysis and (2) the methods for clip verification. For more details, please refer to Appendix (Section C).

## 7 Conclusion

This paper presents a newly collected million-scale dataset for text-to-video generation with a focus on user needs. Beyond this, our dataset exhibits minimal overlap with existing datasets and is released under a more permissive license (*i.e.*, CC BY). We first describe the curation process, which involves analyzing user-focused topics, searching for these topics, segmenting the resulting videos, and applying various post-processing techniques. Then, we highlight the differences between our dataset and existing ones in terms of fundamental attributes and topics coverage. Finally, we build a new benchmark to evaluate text-to-video models on user-focused topics and demonstrate that our dataset helps improve model performance in this regard. We encourage both the research community and industry to use our dataset to further advance the field of text-to-video generation.

## Acknowledgments

We sincerely thank OpenAI for their support through the Researcher Access Program. Without their generous contribution, this work would not have been possible.

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

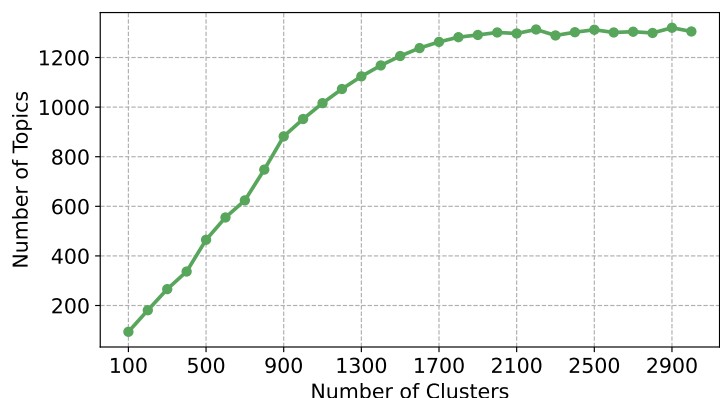

Figure 9: The relationship between pre-set number of clusters for K-means and final number of user-focused topics.

Table 2: The methods for clip verification. We use $1,000$ clips for this ablation study. "Direct GPT-4o" refers to processing videos directly with GPT-4o, and "Overlap" measures the extent to which the predictions of other methods match those of "Direct GPT-4o".

|  | Method | # Verified | Overlap | API Cost |
|---|---|---|---|---|
|  | Direct GPT-4o | 507 | — | $162.23 |
| Brief | GPT-4o | 174 | 0.621 | $0.04 |
|  | GPT-4o-mini | 276 | 0.657 | $0.01 |
| Detail | GPT-4o | 299 | 0.770 | $0.46 |
|  | GPT-4o-mini | 393 | 0.806 | $0.03 |

## A  The Prompt for Concluding Topics

> **Prompt:** Could you describe the topic in the following short sentences using only 1-2 words? Please return only the topic (1-2 word) as a singular noun or in the base form of the verb.

## B  The Prompt for Describing Videos

> **Prompt:** Please describe the content of this video in as much detail as possible, including the objects, scenery, animals, characters, and camera movements within the video. Do not include \n in your response. Start the description with the video content directly. Describe the content of the video and the changes that occur, in chronological order.

## C  Ablation Studies on Curation of VideoUFO

This section performs ablation studies to investigate key components in curating VideoUFO, *i.e.*, examining (1) the impact of the number of clusters used in user-focused topic analysis and (2) the methods used for clip verification.

**The number of clusters.** As shown in Fig. 9, we vary the number of pre-set clusters from $100$ to $3,000$ when analyzing the focused topics of text-to-video users. We observe that initially, as the number of pre-set clusters increases, the number of resulting topics also increases; however, after reaching approximately $2,000$ clusters, this increase stops and begins to fluctuate. Therefore, we finally choose $2,000$ as the number of pre-set clusters. This is reasonable because the number of user-focused topics is limited, and an excessive number of pre-set clusters would eventually merge together. Beyond this, we also experiment with DBSCAN and HDBSCAN [47] to automatically determine the number of clusters; however, we do not obtain reasonable results.

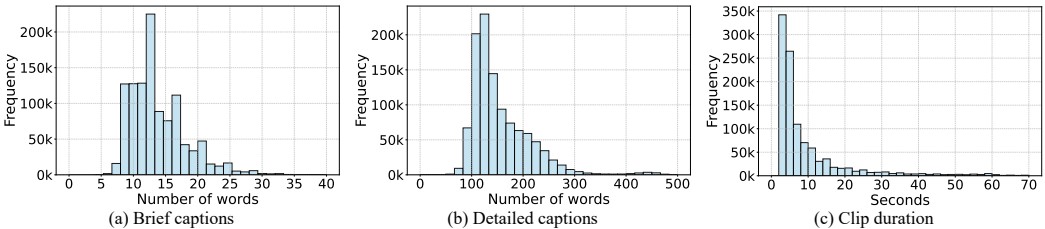

| (a) Brief captions | (b) Detailed captions | (c) Clip duration |

Figure 10: The statistical information of captions and video clips in the proposed VideoUFO dataset. The average word count for brief and detailed captions is $13.8$ and $155.5$, respectively, while the average clip duration is $12.6$ seconds.

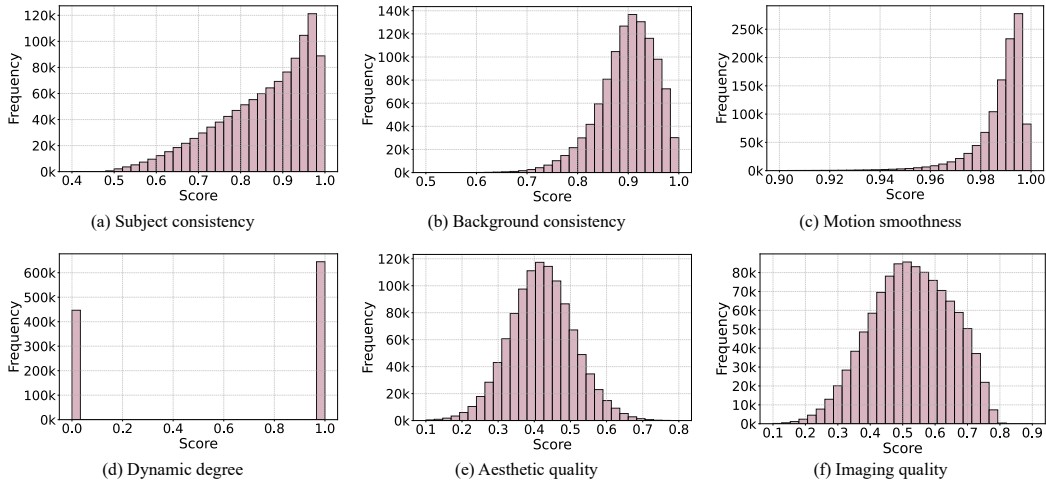

| (a) Subject consistency | (b) Background consistency | (c) Motion smoothness |
| (d) Dynamic degree | (e) Aesthetic quality | (f) Imaging quality |

Figure 11: The statistical distributions of scores assessed by these six video quality metrics, which have been aligned with human perception. Researchers can use these scores to filter videos according to their specific training needs.

**The method for clip verification.** We randomly select $1,000$ video clips to conduct ablation studies on the methods used to verify whether a clip contains a specific topic. Feeding the entire video into GPT-4o (denoted as "Direct GPT-4o") is considered the most accurate approach, and we compare the overlap of other cost-effective methods with it. As shown in Table 2, we observe that: **(1)** Feeding videos directly into GPT-4o is very expensive – processing $1,000$ clips costs $\$162.23$. **(2)** Using detailed captions instead of video clips reduces the API cost by approximately $5,400\times$, while maintaining about $80\%$ prediction overlap. **(3)** Although using brief captions is even cheaper, the overlap drops by $14.9\%$, which is reasonable since a brief caption cannot capture all the information in a clip. **(4)** Interestingly, we find that GPT-4o-mini achieves slightly better performance than GPT-4o. **In conclusion**, for verifying clips, we choose detailed captions with GPT-4o-mini.

## D    The Distributions of Caption Lengths and Clip Durations

Fig. 10 presents the statistical distributions of caption lengths (for both brief and detailed captions) and clip durations.

## E    The Distributions of Video Quality Score

Fig. 11 shows the statistical distributions of scores evaluated in terms of *subject consistency*, *background consistency*, *motion smoothness*, *dynamic degree*, *aesthetic quality*, and *imaging quality*.

## F    Broader Impacts

The open, topic-rich dataset lowers the barrier for global researchers to experiment with text-to-video generation. Its accessibility can spark new tools for education, creativity, and assistive technologies.

