# OpenReview forum: "VideoUFO: A Million-Scale User-Focused Dataset for Text-to-Video Generation"
_NeurIPS.cc/2025/Datasets_and_Benchmarks_Track — NeurIPS 2025 Datasets and Benchmarks Track poster_

### Official Review · Reviewer_LCkh · 2025-06-23

**Ethics Flags:** Data privacy, copyright, and consent,…
**Rating:** 5
**Confidence:** 5

**Summary:**

This paper introduces the VideoUFO dataset, which aims to improve text-to-video models by focusing on user-specific topics. The VideoUFO dataset contains over 1.09 million video clips sourced from YouTube under the CCBY4.0 license (1291 user-focused topics), with minimal overlap with existing datasets. Experiments show that current models perform poorly when handling user-centric topics, while simple models trained on VideoUFO outperform other models when tackling challenging topics.

**Additional Feedback:**

See the comments above.

**Dataset Code Accessibility:**

Yes

**Dataset Code Comments:**

Code: https://github.com/WangWenhao0716/BenchUFO
Dataset: https://huggingface.co/datasets/WenhaoWang/VideoUFO

**Ethical Considerations:**

No, there are no or only very minor ethics concerns

**Final Justification:**

The authors’ rebuttal has addressed most of my concerns.


**For Q1**: The categories used in VideoUFO are largely inherited from VidProM, which has already conducted a thorough exploration of category selection. Therefore, I consider this issue resolved.

**For Q2**: The authors claim in the paper that the video duplication rate does not exceed 0.29%. I believe this low rate results from their careful curation of the dataset. Upon my request, they compared ChronoMagic-Pro with OpenHumanVid and found a duplication rate of 49%. Although the result is not very promising, I appreciate the authors’ honesty and transparency. Hence, I consider this issue partially resolved.

**For Q3**: Following my suggestion, the authors added more references to past NeurIPS D&B papers to make the paper feel more grounded and complete. Therefore, I consider this issue resolved.

**For Q4**: I’ve always been somewhat confused about the evaluation criteria for the D&B Track—some say novelty is required, while others emphasize practical contributions to the community. Although this work largely follows existing methodologies, the authors have provided a reasonable justification. Therefore, I consider this issue partially resolved.

**Limitations Weaknesses:**

W1. 1291 categories are extracted from VidProM, the reliability of the categories depends on the reliability of the previous work. Is there a better way to get the categories?

W2. The authors compared the YouTube IDs of the downloaded videos with those in various existing datasets and found only a 0.29% overlap. While these datasets are also designed for text-to-video generation, they are all open-domain and differ from VideoUFO in focus. It may be worth supplementing the comparison with more targeted datasets such as ChronoMagic-Pro, which focuses on time-lapse videos, and OpenHumanVid, which centers on the human domain—both of which, like VideoUFO, involve purposefully curated data.

W3. In my opinion, the article by D&B should include a comprehensive literature review (with extensive references), but currently, it only contains 46 references.

W4. The dataset and benchmark processing largely follow existing methodologies, with the main novelty lying in the way categories are constructed.

**Strengths Contributions:**

S1. Unlike previous video datasets that collect videos without a clear purpose, VideoUFO identifies 1,291 categories based on users’ real-world interests and curates 1.09 million high-quality video clips. This targeted approach addresses the low user satisfaction commonly seen in current video generation models.

S2. Only 1,675 video IDs in VideoUFO (0.29%) overlap with existing datasets, indicating that it contains a large amount of novel content valuable to the research community. Admittedly, while the dataset comparison is not exhaustive, it nonetheless provides a reasonable reflection of the current landscape.

S3. Fine-tuning existing video generation models with VideoUFO significantly improves performance on the most challenging user-focused topics, whereas current text-to-video foundation models still struggle with these topics.

S4. The content is very substantial and the reading is smooth.

---

> ### Author Rebuttal · Authors · 2025-07-30
>
> We sincerely thank you for your positive, thoughtful, and helpful feedback. We will incorporate all your suggestions into the final version of our paper. We are encouraged that you found *“Unlike previous video datasets that collect videos without a clear purpose, VideoUFO identifies 1,291 categories based on **users’ real-world interests**”*, *“Only 1,675 video IDs in VideoUFO (0.29%) overlap with existing datasets, indicating that it contains a large amount of **novel content valuable** to the research community”*, *“Fine-tuning existing video generation models with VideoUFO **significantly** improves performance on the most challenging user-focused topics”*, and *“The content is very **substantial** and the reading is **smooth**”*. We address your questions below.
>
> **Q1. 1291 categories are extracted from VidProM, the reliability of the categories depends on the reliability of the previous work. Is there a better way to get the categories?**
>
> A1. Thank you for the insightful question. Our conclusion is that **VidProM is currently the most suitable choice for this research.** The reasons are detailed below:
>
> **1. Most other video generation products only make a small portion of the user-provided prompts publicly available, and thus cannot support our research well.** VidProM is a dataset composed of **million-scale real-user prompts from a mainstream T2V model, i.e., Pika**. We also explore the user-provided prompts and galleries of 5 mainstream T2V products in addition to Pika. The investigated products and the approximate number of their publicly available videos are detailed below. The scale of prompts provided by these video generation products is relatively small and cannot support our research.
> |Runway Watch |Veo 3 Showcase|Sora Gallery|Hailuo Demo|Genmo Discover|
> |-|-|-|-|-|
> |205|9|35|30|2000|
>
> **2. Some community galleries, such as Krea Gallery, are large in scale but still not suitable for this research.** Specifically, a **major issue** is that these galleries tend to feature **only aesthetically pleasing (successful) results**, while **discarding failed generations**. This issue actually **hinders our research**, as we are particularly interested in prompts (topics) that *people focus on but current models **fail** to generate well*. In contrast, the VidProM includes ***all*** user-provided prompts and the generated videos from Pika within a given period.
>
>
> **Q2. The authors compared the YouTube IDs of the downloaded videos with those in various existing datasets and found only a 0.29% overlap. It may be worth supplementing the comparison with more targeted datasets such as ChronoMagic-Pro, which focuses on time-lapse videos, and OpenHumanVid, which centers on the human domain.**
>
> A2. Thank you for the constructive suggestion. Following it, we compare our VideoUFO with ChronoMagic-Pro and OpenHumanVid below. **We will cite these two impressive works and include this comparison in the final version.**
> |Dataset|#Videos|Avg. Lengths of Videos|Avg. Words of Captions|Resolution|Domain|#Topic|License|
> |-|-|-|-|-|-|-|-|
> | ChronoMagic-Pro |460K|234s|1794.0|720p|Time-lapse|792|CC BY|
> | OpenHumanVid |13.2M|4.6s|20~30|720p+|Human|-|CC BY-NC-ND|
> | VideoUFO |1M |12.6s |155.5|720p|Users|1,291|CC BY|
>
> Besides the basic information above, we observe:
>
> 1. The dataset ChronoMagic-Pro, which focuses on time-lapse videos, contains **792** topics, **633** of which are user-focused. These account for approximately **49%** of the **1,291 user-focused topics** in our VideoUFO dataset. Additionally, **only 102 videos overlap between VideoUFO and ChronoMagic-Pro.**
>
> 2. Although OpenHumanVid, which specializes in the human domain, features approximately 10 times more video clips, it has the following limitations: (1) the average video length is relatively **short**; (2) the captions **lack sufficient detail**; and (3) it adopts **a more restrictive license (CC BY-NC-ND)**.
>
> *Note: Access to OpenHumanVid requires approval. Although we have submitted the application form and emailed the owner, we have not received a response so far. As a result, we are currently unable to determine the number of topics in the dataset and the overlap between it with our VideoUFO. However, we will update this information in our paper once access is granted.*
>
> **Q3. In my opinion, the article by D&B should include a comprehensive literature review (with extensive references), but currently, it only contains 46 references.**
>
> A3. Thank you for your question. Based on our understanding of the dataset and the benchmark track, there might be a slight difference. Admittedly, many D&B papers include comprehensive literature reviews or extensive references. However, we find that most of them primarily focus on either (1) **building comprehensive benchmarks** to evaluate a large number of methods (e.g., ChronoMagic-Bench [1]), or (2) constructing datasets that **integrate numerous existing datasets**  (e.g. Terra [2]) or are generated **using various established methods** (e.g., DF40 [3]). In contrast, datasets collected **directly from the web or other sources** typically include fewer references (e.g., Openmathinstruct-1 [4] and LVD-2M [5], and **our VideoUFO**).
>
> **Nevertheless, following your suggestion, we would be happy to include additional references.** Specifically, (1) we will add both classical and recent text-to-video datasets **[6-18]** to the Related Works section (Text-Video Datasets), and (2) we will include recent text-to-video methods, such as **Vidu**, **Wan2.2**, **Seedance**, **StepVideo**, **IPOC**, **MiracleVision**, **CausVid**, **Luma**, **MiniMax-Video**, **STIV**, **AccVideo**, **Gen-3**, and **Jimeng**, in the Related Works section (Text-to-Video Generation).
>
> [1] Yuan, Shenghai, et al. "Chronomagic-bench: A benchmark for metamorphic evaluation of text-to-time-lapse video generation." Advances in Neural Information Processing Systems (Datasets and Benchmarks Track) 37 (2024): 21236-21270.
>
> [2] Chen, Wei, et al. "Terra: A multimodal spatio-temporal dataset spanning the earth." Advances in Neural Information Processing Systems 37 (Datasets and Benchmarks Track) (2024): 66329-66356.
>
> [3] Yan, Zhiyuan, et al. "Df40: Toward next-generation deepfake detection." Advances in Neural Information Processing Systems 37 (Datasets and Benchmarks Track) (2024): 29387-29434.
>
> [4] Toshniwal, Shubham, et al. "Openmathinstruct-1: A 1.8 million math instruction tuning dataset." Advances in Neural Information Processing Systems 37 (Datasets and Benchmarks Track) (2024): 34737-34774.
>
> [5] Xiong, Tianwei, et al. "Lvd-2m: A long-take video dataset with temporally dense captions." Advances in Neural Information Processing Systems 37 (Datasets and Benchmarks Track) (2024): 16623-16644.
>
> [6] Howto100m: Learning a text-video embedding by watching hundred million narrated video clips
>
> [7] Acav100m: Automatic curation of large-scale datasets for audio-visual video representation learning
>
> [8] Merlot: Multimodal neural script knowledge models
>
> [9] Vatex: A large-scale, high-quality multilingual dataset for video-and-language research
>
> [10] Towards automatic learning of procedures from web instructional videos
>
> [11] Activitynet: A large-scale video benchmark for human activity understanding
>
> [12] Opens2v-nexus: A detailed benchmark and million-scale dataset for subject-to-video generation
>
> [13] Chronomagic-bench: A benchmark for metamorphic evaluation of text-to-time-lapse video generation
>
> [14] Vript: A video is worth thousands of words
>
> [15] Hoigen-1m: A large-scale dataset for human-object interaction video generation
>
> [16] Wisa: World simulator assistant for physics-aware text-to-video generation
>
> [17] UltraVideo: High-Quality UHD Video Dataset with Comprehensive Captions
>
> [18] MedGen: Unlocking Medical Video Generation by Scaling Granularly-annotated Medical Videos
>
> **Q4. The dataset and benchmark processing largely follow existing methodologies, with the main novelty lying in the way categories are constructed.**
>
> A4. Thank you for this insightful question. We respectfully clarify that our use of established data processing methodologies was a **deliberate strategic choice**, *not a lack of novelty*. This choice allows us to focus our innovation on what we believe are pressing gaps in the text-to-video generation. Specifically, we frame our novelty in two key areas:
>
> **1. Resource-level innovation, i.e., providing a user-focused, unique, and compliant data.** Unlike recent datasets (e.g., Panda-70M, LVD-2M, Koala-36M, VidGen-1M, and OpenVid-1M) that are with open domains and heavily sourced from HD-VILA-100M, VideoUFO centers on **user-focused topics** and has a minimal **0.29% overlap** with existing public datasets. This ensures that models can gain **user-focused new knowledge**. Furthermore, by using YouTube's official API and Creative Commons licenses, we address the **compliance issues**, offering a flexible resource for both academic and commercial use.
>
> **2. Evaluation-level innovation, i.e., constructing a user-centric benchmark**. We introduce the first benchmark designed to evaluate text-to-video models on **user-focused topics**, complementing existing metrics which focus on semantic consistency and visual qualities. We observe that the current 16 text-to-video models have some poor-performing topics, and (2) our model achieves the highest similarities on the worst-performing topics while maintaining performance on the best-performing ones.

---

> > ### Comment · Reviewer_LCkh · 2025-08-01
> > **Official Comment by Reviewer LCkh**
> >
> > Thanks for the authors' rebuttal. My concerns are solved. I keep my acceptance recommendation. Lastly, I hope VideoUFO will make a valuable contribution to the video generation community.

---

> > > ### Author Response · Authors · 2025-08-01
> > >
> > > We sincerely thank you again for your review and are happy to see that the concerns you raised have been resolved.

---

### Official Review · Reviewer_wJp8 · 2025-06-25

**Rating:** 5
**Confidence:** 4

**Summary:**

VideoUFO is a million-scale video dataset for text-to-image video generative models, especially focusing on the topics preferred by users for video generation. This dataset has several attractive characteristics, such as 1) this dataset has minimal overlap with previous datasets and 2) its curated topics and video quality, and 3) the flexible licenses of collected videos. Furthermore, they propose benchUFO for evaluating the T2V generative models. They select the topics of concrete nouns for resolving the ambiguity of generated videos and create a description by providing the generated videos to video understanding models. Then, they measure the similarity between an input prompt and the created description. They validate that the model trained by the proposed dataset achieves higher consistency on an input prompt and generated video, furthermore reducing the quality gap between videos of best and worst performing topics.

**Dataset Code Accessibility:**

Yes

**Dataset Code Comments:**

They release VideoUFO on Huggingface, which is a widely used platform to share datasets.

**Ethical Considerations:**

No, there are no or only very minor ethics concerns

**Final Justification:**

Authors rebuttal resolves my issue, and I believe the large scale video dataset will contribute the research field of video generative model significantly.

Thus I keep my original rating.

**Limitations Weaknesses:**

- One weakness of the paper is that their evaluation was only performed on the proposed VideoUFO dataset. It would be great if there were evaluations on different datasets with the MVDiT trained on VideoUFO. Also, it only validates the consistency in the video prompt, without consideration of the quality of the generated videos. However, I understand that the limited computational resource makes the baseline hard to achieve a quality comparable to the state-of-the-art video generative models.

**Strengths Contributions:**

- The proposed VideoUFO datasets have several advantages as a dataset. For example, 1) it is a million-scale video dataset with minimal overlap (0.29%) to the videos from previous datasets and 2) it contains videos with curated topics and quality, and 3)a  flexible license, expected to encourage the application of this dataset.

- They evaluate the various types of T2V generative models on the proposed benchUFO, and validate that the model trained on VideoUFO achieves the highest consistency on the given prompt with reduced gap between the video quality generated by different topics.

---

> ### Author Rebuttal · Authors · 2025-07-30
>
> We sincerely thank you for your positive, thoughtful, and helpful feedback. We will incorporate all your suggestions into the final version of our paper. We are encouraged that you found *“the proposed VideoUFO (1) is a **million-scale** video dataset with **minimal overlap** with others, (2) contains videos with **curated topics** and quality, and (3) is with a **flexible license**”*, and *“authors evaluate the various types of T2V generative models on the proposed benchUFO, and validate that the model trained on VideoUFO achieves the **highest** consistency”*. We address your questions below.
>
> **Q1. One weakness of the paper is that their evaluation was only performed on the proposed VideoUFO dataset. It would be great if there were evaluations on different datasets with the MVDiT trained on VideoUFO. Also, it only validates the consistency in the video prompt, without consideration of the quality of the generated videos. However, I understand that the limited computational resource makes the baseline hard to achieve a quality comparable to the state-of-the-art video generative models.**
>
> A1. Thank you for the insightful question. As you said, to begin with, we would like to acknowledge that our trained model **solely** on VideoUFO is unlikely to achieve state-of-the-art on standard benchmarks due to limitations in *model size*, *training data and methods*, and *computational resources*. **Nevetheless, we try to address your concern from three aspects:**
>
> **1. Finetuning or distilling on our VideoUFO preserves the performance of state-of-the-art video generative models on the standard benchmark, VBench.** This experiment serves as a complement by providing evaluations on other datasets, mitigating the limitation of our previous focus on the VideoUFO. In a recent work [1] from **Meta**, they chose to use *VideoUFO* over other popular datasets such as *WebVid*, *HD-VILA*, *InternVid*, and *MiraData* (see Section 4.6.1, Dataset). Table 4 in that paper shows that, after distilling on our dataset, ```Wan2.1-LinGen``` achieves performance **comparable** to (some even better than) the state-of-the-art ```Wan2.1``` across **all**  metrics in established VBench. For your convenience, we attach their Table 4 below.
>
> |        Metrics   | Wan2.1   | LTX-Video | Wan2.1-LinGen-s1 | Wan2.1-LinGen |
> |--------------------------|----------|-----------|------------------|----------------|
> | Object Class             | 92.93%   | 76.91%    | 90.41%           | 91.88%         |
> | Multiple Objects         | 84.30%   | 28.25%    | 83.55%           | 84.22%         |
> | Human Action             | 95.80%   | 89.20%    | 95.57%           | 96.20%         |
> | Color                    | 85.63%   | 77.17%    | 83.89%           | 84.47%         |
> | Spatial Relationship     | 79.33%   | 46.86%    | 77.91%           | 78.21%         |
> | Scene                    | 55.48%   | 29.46%    | 54.44%           | 54.93%         |
> | Appearance Style         | 21.35%   | 19.61%    | 21.42%           | 21.82%         |
> | Temporal Style           | 25.24%   | 21.36%    | 25.34%           | 25.51%         |
> | Overall Consistency      | 27.22%   | 23.26%    | 26.01%           | 26.75%         |
> | **Semantic Score**           | 80.48%   | 60.60%    | 79.28%           | 80.19%         |
> | Subject Consistency      | 93.02%   | 90.48%    | 92.33%           | 92.52%         |
> | Background Consistency   | 96.73%   | 93.34%    | 96.32%           | 96.49%         |
> | Temporal Flickering      | 98.61%   | 98.34%    | 98.57%           | 98.68%         |
> | Motion Smoothness        | 97.64%   | 99.13%    | 97.72%           | 98.25%         |
> | Aesthetic Quality        | 66.96%   | 58.03%    | 65.79%           | 66.23%         |
> | Imaging Quality          | 66.81%   | 59.21%    | 66.21%           | 66.42%         |
> | Dynamic Degree           | 80.56%   | 85.28%    | 79.83%           | 80.21%         |
> | **Quality Score**            | 84.69%   | 82.02%    | 84.17%           | 84.70%         |
> |**Total Score**             | 83.84%   | 77.73%    | 83.20%           | 83.80%         |
>
>
> [1] Wang, Hongjie, et al. "LinGen-Uni: A Universal Linear-Complexity Framework for High-Resolution Minute-Length Text-to-Video Generation." (2025). Preprint Available at ResearchSquare.
>
>
> **2. Evaluated by established metrics, VideoUFO maintains visual quality on par with similar-scale datasets.** We compare the visual quality of models trained using *VideoUFO*, *OpenVid*, and *VidGen*. We use the prompts from our BenchUFO benchmark and six established metrics from standard VBench. The experimental results are as detailed below. Note that OpenVid and VidGen were both specifically designed to improve visual quality.
> |Models|Subject consistency|Background consistency|Motion smoothness|Dynamic degree| Aesthetic quality|Imaging quality|
> |-|-|-|-|-|-|-|
> |MVDiT-VidGen|0.941|0.954|0.970|0.328|0.513|0.570|
> |MVDiT-OpenVid|0.976|0.973|0.981|0.288|0.561|0.614|
> |MVDiT-VideoUFO|0.964|0.967|0.974|0.301|0.545|0.588|
>
> **3. We are in discussions with leading companies, including Alibaba, Baidu, and Tencent, to scale up our VideoUFO.** We also acknowledge that, without broader collaboration and resources beyond VideoUFO, it is **impossible** to reach the level of video generation quality required for commercial applications. Nevertheless, multiple companies have recognized the potential of our **data collection methodology** and are considering using the dataset or leveraging the approach to construct larger-scale datasets to enhance their state-of-the-art models.

---

### Official Review · Reviewer_mdca · 2025-06-28

**Rating:** 5
**Confidence:** 5

**Summary:**

This paper introduces a new large-scale video dataset, comprising one million videos, designed for tasks in both video generation and video understanding.

**Dataset Code Accessibility:**

Yes

**Dataset Code Comments:**

Yes. I have carefully checked the dataset on HuggingFace.

**Ethical Considerations:**

No, there are no or only very minor ethics concerns

**Final Justification:**

I carefully checked the Gemini version of the caption and found it to be much better, so I decided to increase my score.

**Limitations Weaknesses:**

1.  The claim that the dataset is "collected in line with real text-to-video users’ focus or preferences" is questionable. The authors define "user's focused topics" solely based on the VidProM dataset. This approach lacks rigor, as VidProM is known to be a noisy dataset and may not accurately reflect genuine user interests, which also evolve over time. A more sound approach would be to analyze user-generated content from galleries of mainstream T2V products to capture more authentic user prompts.
2.  The dataset's captions were generated using Qwen2-VL-7B. However, as noted in prior work (e.g., Tarsier), the Qwen2/2.5-VL series is prone to producing hallucinations and omissions, which is a critical flaw for T2V applications. I have carefully inspected the captions in this dataset and confirmed that the description quality is relatively poor.
3.  The justification for not evaluating on established benchmarks is unconvincing (Line 297). The authors' methodology involves creating a new data classification system, collecting data tailored to it, and then demonstrating their model's superiority within this same self-defined framework. This circular evaluation fails to provide a broad assessment. T2V evaluation encompasses a wide range of objectives, and the proposed system represents only a narrow perspective. The absence of comparisons against standard benchmarks significantly weakens the paper's claims.
4.  The experiment in Figure 7 provides limited insight. To demonstrate the value of their data collection strategy, a more appropriate baseline would be a model trained on a 1M subset of a larger dataset like Koala-36M, rather than comparing against OpenVid and VidGen. Additionally, including results from leading closed-source commercial T2V models would help establish a potential upper bound for the proposed evaluation metric. As presented, the results do not offer a compelling demonstration of the value of the authors' noun/concept-filtering methodology.
5.  The experiments in Figure 7 are limited to a single architecture (MVDiT). The conclusions would be more robust if the authors had demonstrated the dataset's effectiveness by fine-tuning a pre-existing, state-of-the-art T2V model.
6.  The qualitative comparison in Figure 8 is flawed. The authors appear to have cherry-picked different prompts for each model, which prevents a fair and direct comparison of their capabilities.

**Strengths Contributions:**

1.  The dataset consists of newly collected videos with minimal overlap with existing public datasets, making it a valuable addition to the open-source community.
2.  The dataset's construction is tailored for T2V generation, with a particular emphasis on expanding the diversity of noun concepts.

---

> ### Author Rebuttal · Authors · 2025-07-30
>
> We sincerely thank you for your positive, thoughtful, and helpful feedback. We will incorporate all your suggestions into the final version. We are encouraged that you found *“the dataset is with **minimal overlap** with existing datasets, making it a **valuable** addition”*, and *“the dataset expands the **diversity** of noun concepts”*. We address your questions below.
>
> **Q1. The claim that the dataset is "collected in line with real text-to-video users’ focus or preferences" is questionable. A more sound approach would be to analyze user-generated content from galleries of mainstream T2V products.**
>
> A1. We apologize for the confusion. To begin with, we would like to clarify that VidProM is already a dataset composed of **million-scale real-user prompts and galleries from a mainstream T2V model, i.e., Pika**. While we acknowledge the noise in VidProM, we mitigate this by using GPT-4o to conclude topics. Our manual inspection finds that GPT-4o can accurately extract topics from prompts with noise.
>
> Following your suggestions, we investigate the galleries of mainstream T2V products beyond Pika (VidProM). However, we identify **two issues** that prevent us from using their galleries.
>
> **1. Most other video generation products only make a small portion of the user-provided prompts publicly available, and thus cannot support our research well.** Beyond VidProM (Pika), we also explore the user-provided prompts and galleries of 5 T2V products. The products and the approximate number of their publicly available videos are detailed below. The scale of prompts provided by these products is relatively small and cannot support our research.
> |Runway Watch |Veo 3 Showcase|Sora Gallery|Hailuo Demo|Genmo Discover|
> |-|-|-|-|-|
> |205|9|35|30|2000|
>
> **2. Some community galleries, such as Krea Gallery, are large in scale but still not suitable for this research.** Specifically, a **major issue** is that these galleries tend to feature **only aesthetically pleasing (successful) results**, while **discarding failed generations**. This issue actually **hinders our research**, as we are particularly interested in prompts (topics) that *people focus on but current models **fail** to generate well*. In contrast, the VidProM includes ***all*** user-provided prompts and the generated videos from Pika within a given period.
>
> In conclusion, while we acknowledge that user interests may evolve over time, **VidProM remains the most suitable choice at present.** In the future, if more suitable galleries or datasets become available, we will update VideoUFO accordingly.
>
> **Q2. The dataset's captions were generated using Qwen2-VL-7B. However, as noted in prior work, the Qwen2/2.5-VL series is prone to producing hallucinations and omissions.**
>
> A2. Thank you for the constructive suggestion. We selected `Qwen2-VL-7B` as it was the best video understanding model that could be run with academic-level computing resources when constructing VideoUFO. Nevertheless, **we agree with your opinion**, and in response, we **re-captioned all video clips** using the best video understanding model, **`Gemini-2.5-pro`**.  This labeling effort consumed **$12,691.95** of our API funding and successfully produced **1,091,650** captions out of **1,091,712** clips. We observe a clear improvement in caption accuracy and hope this costly re-labeling benefits the video generation community. Due to NeurIPS policy, we cannot update the new captions to our dataset or provide a direct download link. **Nevertheless, we will release these improved captions to the community after the review process.** Here, we give an example to compare the outputs of `Gemini-2.5-pro` and `Qwen2-VL-7B`.
>
> ID: `1sPMa6fAGX8.11`
>
> `Qwen2-VL-7B` exhibits both *omission* and *hallucination* issues: It omits the **“dark, metallic, armored finger”** and misidentifies it as *“wood or stone”*. Moreover, it hallucinates the visual detail *“The lighting in the scene is warm, casting a golden glow over the ring and the surrounding area”*, which should be **“glow from within, taking on a fiery orange hue”**.
>
> Caption from `Qwen2-VL-7B`
> ```text
> The video begins with a close-up shot of a dark, textured surface, possibly wood or stone, with a ring prominently displayed. … The text appears to be in an ancient or fantasy-style script, possibly Elvish or another fictional language. The lighting in the scene is warm, casting a golden glow over the ring and the surrounding area, …
> ```
>
> Caption from `Gemini-2.5-pro`
> ```text
> The video opens with an extreme close-up on a plain, smooth gold ring worn on a dark, metallic, armored finger. The background is a blurry, warm, golden-brown. The ring begins to glow from within, taking on a fiery orange hue. As the glow intensifies, intricate, fiery script materializes and etches itself across the surface of the band, wrapping around it. The camera remains static, focused on the ring's transformation throughout the clip.
> ```
>
> **Q3. The justification for not evaluating on established benchmarks is unconvincing (Line 297).**
>
> A3. Thank you for the insightful question. We try to address this concern from three aspects:
>
> **1. Evaluated by established metrics, VideoUFO maintains visual quality on par with similar-scale datasets.** We compare the visual quality of models trained using *VideoUFO*, *OpenVid*, and *VidGen*. We use the prompts from our BenchUFO and six established metrics from VBench. The experiments are as detailed below. Note that OpenVid and VidGen were both designed to improve visual quality.
> |Models|Subject consistency|Background consistency|Motion smoothness|Dynamic degree| Aesthetic quality|Imaging quality|
> |-|-|-|-|-|-|-|
> |MVDiT-VidGen|0.941|0.954|0.970|0.328|0.513|0.570|
> |MVDiT-OpenVid|0.976|0.973|0.981|0.288|0.561|0.614|
> |MVDiT-VideoUFO|0.964|0.967|0.974|0.301|0.545|0.588|
>
> **2. Distilling on VideoUFO preserves the performance of state-of-the-art models on the established benchmark, VBench.** In a recent work [1] from **Meta**, they chose to use *VideoUFO* over other popular datasets such as *WebVid*, *HD-VILA*, *InternVid*, and *MiraData* (see Section 4.6.1, Dataset). Table 4 in that paper shows, after distilling on our dataset, `Wan2.1-LinGen` achieves performance **comparable** to (even better than) the state-of-the-art `Wan2.1` across **all**  metrics in VBench. For your convenience, we attach their Table 4 below.
> |Metrics|Wan2.1|Wan2.1-LinGen|
> |-|-|-|
> |Object Class|92.93%|91.88%|
> |Multiple Objects|84.30%|84.22%|
> |Human Action|95.80%|96.20%|
> |Color|85.63%|84.47%|
> |Spatial Relationship|79.33%|78.21%|
> |Scene|55.48%|54.93%|
> |Appearance Style|21.35%|21.82%|
> |Temporal Style|25.24%|25.51%|
> |Overall Consistency|27.22%|26.75%|
> |**Semantic Score**|80.48%|80.19%|
> |Subject Consistency|93.02%|92.52%|
> |Background Consistency|96.73%|96.49%|
> |Temporal Flickering|98.61%|98.68%|
> |Motion Smoothness|97.64%|98.25%|
> |Aesthetic Quality|66.96%|66.23%|
> |Imaging Quality|66.81%|66.42%|
> |Dynamic Degree|80.56%|80.21%|
> |**Quality Score**|84.69%|84.70%|
> |**Total Score**|83.84%|83.80%|
>
> [1] Wang, Hongjie, et al. "LinGen-Uni: A Universal Linear-Complexity Framework for High-Resolution Minute-Length Text-to-Video Generation." (2025). Preprint Available at ResearchSquare.
>
> **3. We are in discussions with leading companies, including Alibaba, Baidu, and Tencent, to scale up VideoUFO.** We acknowledge that our trained model **solely** on VideoUFO is unlikely to achieve state-of-the-art on standard benchmarks due to limitations in *model size*, *training data and methods*, and *computational resources*. **We kindly ask for your understanding.** Nevertheless, multiple companies have recognized the potential of our data collection methods and are considering using the dataset or the approach to construct larger-scale datasets to enhance their models.
>
> **Q4. A more appropriate baseline would be a model trained on a 1M subset of a larger dataset rather than OpenVid and VidGen. Additionally, including results from leading closed-source commercial T2V models would help establish an upper bound.**
>
> A4. We apologize for any confusion. First, we would like to clarify that the OpenVid and VidGen are **already** 1M subsets of the larger datasets, HD-VILA-100M/Panda-70M. Therefore, to show the effectiveness of our data collection strategy, we compare it against these two datasets.
>
> Secondly, we agree that including results from leading closed-source commercial T2V models is beneficial. Therefore, we included the **Veo 3** in our benchmark, which incurred an API cost of  **$30,072.00**. Experiments below indicate that Veo 3 sets a new state-of-the-art.
> |Models|Low 10|Low 50|Top 50|Top 10|
> |-|-|-|-|-|
> |Veo 3|0.492|0.525|0.651|0.678|
>
> **Q5. The experiments in Fig. 7 are limited to a single architecture. The authors should demonstrate the dataset's effectiveness by fine-tuning a pre-existing, state-of-the-art T2V model.**
>
> A5. Thank you for the constructive suggestion. Following it, we finetune `Wan2.1-T2V-1.3B-Diffusers` on our VideoUFO and observe performance improvement on the worst-performing topics. We kindly ask for your understanding in our choice of a relatively small model due to limited rebuttal time and computing resources. The experiments are detailed below.
> |Models|Low 10|Low 50|Top 50|Top 10|
> |-|-|-|-|-|
> |w/o finetune|0.340|0.376|0.585|0.624|
> |w finetune|0.396|0.425| 0.597 |0.635|
>
> **Q6. The qualitative comparison in Fig. 8 is flawed. The authors appear to have cherry-picked different prompts for each model.**
>
> A6. We apologize for any confusion. We use a diverse set of prompts to demonstrate that **our model trained on VideoUFO exhibits consistently strong performance across a wide range of topics**. In fact, we observe that most existing models struggle to produce high-quality results for topics such as *Aztec warriors*, *Diwali*, *animal cells*, *karate*, and *ancient Egypt*. Following your suggestion, we will also include comparisons using the same prompts in the final version.

---

> > ### Comment · Reviewer_mdca · 2025-08-03
> >
> > I carefully checked the Gemini version of the caption and found it to be much better, so I decided to increase my score.

---

> > > ### Author Response · Authors · 2025-08-03
> > >
> > > Thank you for the positive feedback and for increasing the score. I'm glad to hear you found the Gemini version to be a clear improvement!

---

### Official Review · Reviewer_GvT4 · 2025-07-03

**Rating:** 5
**Confidence:** 4

**Summary:**

This paper proposes VideoUFO, a video dataset that is specifically curated to align with users' focus in real-world scenarios. Videos in this dataset are selected from YouTube and filtered based on the popular topics that users are more interested in. Only a tiny portion of the videos in this dataset overlap with existing video datasets. Topic-specific video generative benchmarking results demonstrate that VideoUFO enhances the text-video alignment of models on the worst-performing topics.

**Dataset Code Accessibility:**

Yes

**Dataset Code Comments:**

VideoUFO is readily accessible via https://huggingface.co/datasets/WenhaoWang/VideoUFO. The format of this dataset is well-explained in the dataset card.

**Ethical Considerations:**

No, there are no or only very minor ethics concerns

**Final Justification:**

Thanks for the authors' detailed rebuttal. It addressed most of my concerns. Thus, I increased my rating and remain positive towards the acceptance of this paper.

**Limitations Weaknesses:**

1. The experimental setting used to demonstrate the effectiveness of VideoUFO does not appear to be fair. The detailed captions in VideoUFO are obtained by employing QWen2-VL-7B for video annotation. In the experimental setting, the same model, QWen2-VL-7B, is used to provide the description of generated videos. The given descriptions are then used to calculate the similarity with the input prompt. Models trained on VideoUFO potentially adapt more to the description style of QWen2-VL-7B, giving them an unfair advantage in this experimental setting. The authors are advised to change the model they use to describe the generated videos in this experiment.
2. The benchmark used to demonstrate the effectiveness of VideoUFO focuses solely on the text-video alignment of generated videos, ignoring the visual quality. The authors are advised to show that VideoUFO does not hurt the visual quality of generated videos compared to existing datasets.
3. This paper indicates that VideoUFO covers more topics than even the combination of all the existing datasets. The authors are advised to provide more examples of the topics currently missing from the existing datasets and their corresponding popularity rankings. How many prompts are related to those missing topics?
4. The current length of videos in VideoUFO is relatively short (mostly less than 10 seconds). The authors are encouraged to integrate more longer videos into VideoUFO in the future.

**Strengths Contributions:**

1. This paper introduces an interesting aspect of measuring the diversity of datasets for text-to-video generation: the number of topics. Extracting topics from user-provided prompts helps the dataset align more closely with users' focus and interests.
2. VideoUFO has a tiny overlap with existing video datasets, introducing new data to expand the scope of open-source video datasets. It is also more diverse than the existing datasets in terms of the number of topics.
3. All videos in VideoUFO are under the Creative Commons license, providing better flexibility for dataset usage.
4. This paper is well-written, organized, and easy to understand.

---

> ### Author Rebuttal · Authors · 2025-07-30
>
> We sincerely thank you for your positive, thoughtful, and helpful feedback. We will incorporate all your suggestions into the final version of our paper. We are encouraged that you found  *“our work introduces an **interesting** aspect”*, *“VideoUFO introduces **new data** and provides **flexibility** for dataset usage”*, and *“the paper is **well-written**, **organized**, and **easy to understand**”*. We address your questions below.
>
> **Q1. The experimental setting used to demonstrate the effectiveness of VideoUFO does not appear to be fair. The authors are advised to change the model they use to describe the generated videos in this experiment.**
>
> A1. Thank you for this insightful suggestion. Based on it, we change the ```QWen2-VL-7B``` to ```gemini-2.5-flash-lite``` and re-describe all the generated videos. We observe that **we still perform better than others, and thus the advantage of models trained on VideoUFO comes from their learned user-focused topics rather than from potential adaptation to the description style**. The detailed experimental results are shown below.
>
> | Models            | Low 10 | Low 50 | Top 50 | Top 10 |
> |-------------------|:------:|:------:|:------:|:------:|
> | Mira              | 0.235  | 0.273  | 0.499  | 0.540  |
> | Show-1            | 0.310  | 0.343 | 0.546  | 0.581  |
> | LTX-Video         | 0.262  | 0.292  | 0.524  | 0.562  |
> | Open-Sora-Plan    | 0.309  | 0.350  | 0.557  | 0.599  |
> | TF-T2V            | 0.327  | 0.352  | 0.562  | 0.609  |
> | Mochi-1           | 0.311  | 0.354  | 0.581  | 0.615  |
> | HiGen             | 0.341  | 0.381  | 0.591  | 0.635  |
> | Open-Sora         | 0.355  | 0.392  | 0.593  | 0.637  |
> | Pika              | 0.363  | 0.394  | 0.587  | 0.621  |
> | RepVideo          | 0.373  | 0.402  | 0.598  | 0.634  |
> | T2V-Zero          | 0.373  | 0.405  | 0.585  | 0.626  |
> | CogVideoX         | 0.361  | 0.402  | 0.603  | 0.634  |
> | Latte-1           | 0.372  | 0.405  | 0.594  | 0.634  |
> | HunyuanVideo      | 0.383  | 0.413  | 0.610  | 0.648  |
> | Lavie             | 0.376  | 0.408  | 0.593  | 0.641  |
> | Pyramidal         | 0.391  | 0.421  | 0.607  | 0.649  |
> | MVDiT-VidGen      | 0.375  | 0.402  | 0.592  | 0.631  |
> | MVDiT-OpenVid     | 0.397  | 0.428  | 0.614  | 0.644  |
> | **MVDiT-VideoUFO**| **0.435** | **0.467** | **0.617** | **0.653** |
>
>
> **Q2. The benchmark used to demonstrate the effectiveness of VideoUFO focuses solely on the text-video alignment of generated videos, ignoring the visual quality. The authors are advised to show that VideoUFO does not hurt the visual quality of generated videos compared to existing datasets.**
>
>
> Thank you for your valuable suggestion. Following it, we compare the visual quality of models trained using *VideoUFO*, *OpenVid*, and *VidGen*. We use the prompts from our BenchUFO benchmark and six visual quality metrics from VBench, *i.e.*, *subject consistency*, *background consistency*, *motion smoothness*, *dynamic degree*,  *aesthetic quality*, and *imaging quality*.
>
> As detailed below, **generally, we achieve performance comparable to OpenVid and outperform VidGen**. Note that OpenVid and VidGen were both specifically designed to improve visual quality.
>
> | Models | Subject consistency | Background consistency | Motion smoothness | Dynamic degree | Aesthetic quality| Imaging quality|
> |------|:------:|:------:|:------:|:------:|:------:|:------:|
> | MVDiT-VidGen      | 0.941  | 0.954  | 0.970  | 0.328  |0.513  |0.570 |
> | MVDiT-OpenVid     | 0.976  | 0.973  | 0.981  | 0.288  |0.561  |0.614 |
> | MVDiT-VideoUFO| 0.964 | 0.967 | 0.974 | 0.301 |0.545 |0.588 |
>
> In addition, researchers from **Meta** also found that our **VideoUFO** dataset **preserves visual quality**: In a recent work [1], they chose to use *VideoUFO* over other popular datasets such as *WebVid*, *HD-VILA*, *InternVid*, and *MiraData* (see Section 4.6.1, Dataset). Table 4 in that paper shows that, after distilling on our dataset, ```Wan2.1-LinGen``` achieves performance **comparable** to (some even better than) the state-of-the-art ```Wan2.1``` across **all** visual quality metrics in VBench-standard. For your convenience, we attach their Table 4 below.
>
> |        Metrics   | Wan2.1   | LTX-Video | Wan2.1-LinGen-s1 | Wan2.1-LinGen |
> |--------------------------|----------|-----------|------------------|----------------|
> | Object Class             | 92.93%   | 76.91%    | 90.41%           | 91.88%         |
> | Multiple Objects         | 84.30%   | 28.25%    | 83.55%           | 84.22%         |
> | Human Action             | 95.80%   | 89.20%    | 95.57%           | 96.20%         |
> | Color                    | 85.63%   | 77.17%    | 83.89%           | 84.47%         |
> | Spatial Relationship     | 79.33%   | 46.86%    | 77.91%           | 78.21%         |
> | Scene                    | 55.48%   | 29.46%    | 54.44%           | 54.93%         |
> | Appearance Style         | 21.35%   | 19.61%    | 21.42%           | 21.82%         |
> | Temporal Style           | 25.24%   | 21.36%    | 25.34%           | 25.51%         |
> | Overall Consistency      | 27.22%   | 23.26%    | 26.01%           | 26.75%         |
> | **Semantic Score**           | 80.48%   | 60.60%    | 79.28%           | 80.19%         |
> | Subject Consistency      | 93.02%   | 90.48%    | 92.33%           | 92.52%         |
> | Background Consistency   | 96.73%   | 93.34%    | 96.32%           | 96.49%         |
> | Temporal Flickering      | 98.61%   | 98.34%    | 98.57%           | 98.68%         |
> | Motion Smoothness        | 97.64%   | 99.13%    | 97.72%           | 98.25%         |
> | Aesthetic Quality        | 66.96%   | 58.03%    | 65.79%           | 66.23%         |
> | Imaging Quality          | 66.81%   | 59.21%    | 66.21%           | 66.42%         |
> | Dynamic Degree           | 80.56%   | 85.28%    | 79.83%           | 80.21%         |
> | **Quality Score**            | 84.69%   | 82.02%    | 84.17%           | 84.70%         |
> |**Total Score**             | 83.84%   | 77.73%    | 83.20%           | 83.80%         |
>
>
> [1] Wang, Hongjie, et al. "LinGen-Uni: A Universal Linear-Complexity Framework for High-Resolution Minute-Length Text-to-Video Generation." (2025). Preprint Available at ResearchSquare.
>
> **Q3. This paper indicates that VideoUFO covers more topics than even the combination of all the existing datasets. The authors are advised to provide more examples of the topics currently missing from the existing datasets and their corresponding popularity rankings. How many prompts are related to those missing topics?**
>
> A3. We apologize for the confusion. In Fig.5, our point is that VideoUFO covers more **user-focused** topics than other datasets, *not* having a wider range of topics. Nevertheless, following your helpful suggestions, we provide five examples of **user-focused** topics missing from each existing dataset. Each triplet in the table below includes (1) a topic name, (2) its popularity ranking, and (3) the number of prompts related to that topic.
>
> | Dataset |  Example 1 |  Example 2    |   Example 3      |    Example 4      |  Example 5 |
> |--------------------------|----------|-----------|--------|----------|----------------|
> |WebVid-10M|(cyberpunk, 8, 8724) |(zombie, 25, 5815)|(astronomy, 59, 3458)|(ufo, 72, 3222)|(gothic, 75, 3191)|
> | HD-VILA-100M| (alien, 2, 11241) | (cyberpunk, 8, 8724) | (explosion, 22, 6184) | (zombie, 25, 5815)|(warrior, 63, 3345) |
> | InternVid | (horror, 1, 23652) |  (cyberpunk, 8, 8724) | (warrior, 63, 3345) |  (ufo, 72, 3222) | (gothic, 75, 3191)|
> | Panda-70M | (alien, 2, 11241) |   (cyberpunk, 8, 8724) | (explosion, 22, 6184) | (zombie, 25, 5815)  | (war, 53, 3695)|
> | LVD-2M | (horror, 1, 23652) |  (cyberpunk, 8, 8724) |    (battle, 11, 8260)      |     (jungle, 54, 3679)    | (ufo, 72, 3222) |
> | MiraData | (alien, 2, 11241) |  (explosion, 22, 6184) |    (robot, 29, 5219)      |    (astronomy, 59, 3458)     | (ufo, 72, 3222)|
> | Koala-36M | (horror, 1, 23652)  |  (alien, 2, 11241)  |    (cyberpunk, 8, 8724)      |    (explosion, 22, 6184)     | (gothic, 75, 3191)|
> | VidGen-1M | (horror, 1, 23652)  |  (cyberpunk, 8, 8724)  |       (war, 53, 3695)    |   (mountain, 61, 3358)      | (ufo, 72, 3222) |
> | OpenVid-1M | (cyberpunk, 8, 8724)  |  (zombie, 25, 5815)  |   (war, 53, 3695)       |    (ufo, 72, 3222)     | (psychedelia, 85, 3073)|
>
> *Note: While we have made an effort to merge topics with similar semantics, it remains possible that an existing dataset may cover a topic similar to one we identified as missing.*
>
>
> **Q4. The current length of videos in VideoUFO is relatively short (mostly less than 10 seconds). The authors are encouraged to integrate more longer videos into VideoUFO in the future.**
>
> A4. Thank you for the constructive suggestion. The current version of VideoUFO follows the well-established pipeline for collecting video clips proposed by Panda-70M, which tends to produce relatively short videos. Based on your feedback, we plan to include longer videos in future versions to further enrich the dataset.

---

> > ### Comment · Reviewer_GvT4 · 2025-08-01
> >
> > Thanks for the authors' detailed rebuttal. It addressed most of my concerns. Thus, I increased my rating and remain positive towards the acceptance of this paper.

---

> > > ### Author Response · Authors · 2025-08-01
> > >
> > > Thank you for your positive feedback and for increasing your rating. We truly appreciate your thoughtful review and support for our work!

---

### Decision · Program_Chairs · 2025-09-18

**Decision:**

Accept (poster)

**Comment:**

This paper proposes the VideoUFO dataset, aiming at enhancing text-to-video generation by focusing on user-specific topics. The authors validated that training a model on the VideoUFO dataset achieves higher consistency on an input prompt and generated video. All four reviewers provided positive feedback on this paper, and during the rebuttal period, the authors further clarified remaining questions. The paper is well-written and the overall methodology is technically sound, as all reviewers have agreed. We believe this paper would be a valuable addition to the conference, thereby recommend accepting this paper.